# Multiple Faces of Cervical Lesions in Children

**DOI:** 10.3390/diagnostics12040792

**Published:** 2022-03-24

**Authors:** Stefana Maria Moisa, Nicolau Andrei, Raluca-Daniela Balcan, Ingrith Miron, Elena Țarcă, Lăcrămioara Butnariu, Elena Cojocaru, Maria Magdalena Leon-Constantin, Cristian Constantin Budacu, Laura Mihaela Trandafir

**Affiliations:** 1Department of Mother and Child Medicine–Pediatrics, “Grigore T. Popa” University of Medicine and Pharmacy, 700115 Iasi, Romania; stefana-maria.moisa@umfiasi.ro (S.M.M.); ingridmiron@hotmail.com (I.M.); laura.trandafir@umfiasi.ro (L.M.T.); 2Department of Dentoalveolar and Maxillofacial Surgery, “Grigore T. Popa” University of Medicine and Pharmacy, 700115 Iasi, Romania; cristibudacu@yahoo.com; 3“Sfanta Maria” Clinical Emergency Hospital, 700115 Iasi, Romania; ralucai26@yahoo.com; 4Department of Surgery II-Pediatric Surgery, “Grigore T. Popa” University of Medicine and Pharmacy, 700115 Iasi, Romania; 5Department of Mother and Child Medicine–Genetics, “Grigore T. Popa” University of Medicine and Pharmacy, 700115 Iasi, Romania; 6Department of Morphofunctional Sciences I—Pathology, “Grigore T. Popa” University of Medicine and Pharmacy, 700115 Iasi, Romania; elena2.cojocaru@umfiasi.ro; 7Medical I Department, “Grigore T. Popa” University of Medicine and Pharmacy, 700115 Iasi, Romania; leon_mariamagdalena@yahoo.com

**Keywords:** adenopathy, sialadenitis, sialolithiasis, child

## Abstract

Pediatric sialolithiasis is a rare condition causing tumefaction, induration, redness, and pain of the affected gland. When the submandibular gland is involved, the lesion can be mistaken for an adenopathy. As there are few studies to elucidate this condition in children, we present a rare case of a 16-year-old female with suggestive symptoms, in which initial clinical examination and two ultrasound examinations mistook the lesion for an adenopathy. A computed tomography examination was performed and the correct diagnosis was established. The patient was sent for oro-maxilo-facial examination and sialolithotomy was performed. A 10-mm yellow calculus was extracted and postoperative case evolution was favorable under wide spectrum antibiotherapy, oral nonsteroidal anti-inflammatory therapy and silagog alimentation. Although submandibular adenopathies are much more frequent in the pediatric age group, when faced with a firm, immobile submandibular lesion, the pediatrician should consider the sialolithiasis diagnosis.

## 1. Introduction

Laterocervical pathology is common during childhood. The pediatrician can have diagnostic dilemmas when facing a child with a tumor-like lesion located on cervical lymph node projection areas, as these lesions may be adenopathies, benign, or malignant tumors or distended, inflamed normal structures. The most frequent cause of palpable adenopathy during childhood is represented by localized or generalized infections. Localized lymphadenopathy represents a volume increase of lymph nodes in neighboring anatomical regions, while generalized lymphadenopathy is defined as lymph node volume increase in more than two non-neighboring regions [1,2,3].

Sialolithiasis is more frequently encountered in patients over the age of 40 and is rarely described in children; some studies did, however, find this condition in children-aged between 4 and 16 years, with an average of 10.4 years, but the percentage of children in those papers did not exceed 3% [4,5].

Escudier et al. [6] described an incidence peak of symptomatic salivary calculi between the ages of 25 and 50. Lustman et al. [7] reported a high incidence in the third and sixth decades of life, while McGurk et al. [8] found a median of 45 years for submandibular gland calculi and 48 years for parotid gland calculi in a series of 455 patients.

Pediatric sialolithiasis exhibits distinct traits, such as smaller dimensions and more distal localization of calculi, as well as shorter symptom duration until diagnosis [9]. This can be due to the fact that children develop symptoms sooner earlier than adults after the calculus is formed [10,11]. Moreover, calculi are more easily formed in adults than children because resting saliva concentrations of calcium and phosphate increase with age [1,11,12], and poor hygiene and unknown metabolic events favor calculus development [13]. Many cases of submandibular gland calculus have been reported, and this can be triggered by salivary properties (pH and calcium concentration) and long curve ductus [14]. Therefore, sialoliths are most frequent in the submandibular glands (80–92%), but can also be encountered in the parotids (6–20%) and sublingual and minor salivary glands (1–2%) [15].

Some studies report a male predominance of the disease [16,17], with males requiring hospitalization twice as frequently as females [18], while others describe equal sex distribution [7], with a positive anamnesis in about 14% of cases [19].

### Justification of the Case Report

Because lithiasic sialadenitis with Wharton duct sialolith is a very uncommon pathological condition in children, the differential diagnosis may be difficult and CT scan investigation is necessary to elucidate this disorder. A rare such case in a pediatric patient is reported here.

## 2. Case Report

A 16-year-old female with no significant prior history was admitted in the General Pediatrics Clinic of “Sfanta Maria” Clinical Emergency Hospital in Iasi, Romania, for evaluation of a right submandibular lesion that appeared 72 h before presentation (Figure 1). The lesion was painful, immobile on the deep and superficial planes, was firm on palpation and caused swallowing difficulties. The rest of the clinical examination was normal and no other adenopathy were palpable.

Routine hematological and biochemical tests did not yield any pathological results. The erythrocyte sedimentation rate was slightly increased, while the C-reactive protein remained within normal limits.

Taking into consideration the submandibular localization of this lesion and the ultrasound scanning performed upon admission that deemed the lesion to be an oval, hypoechogenic adenopathy of 28.2/18.6 mm with a present vascular hilum, that was not accompanied by any left laterocervical adenopathies, we tried to evaluate possible etiologies for this presumed adenopathy. We used immunological methods and excluded Epstein–Barr, Cytomegalovirus, Toxocara, Bartonella, Treponema, and Mycobacterium tuberculosis infections (for this last etiology we also ordered a thoracic X-ray, which was also normal).

In order to explore the other lymph node groups in a non-invasive manner, another lymph node ultrasound examination was performed and all superficial lymph node groups were investigated (laterocervical bilateral, supraclavicular, axillary, retroperitoneal, inguinal). All these lymph node groups were described as normal by this second examiner, who also looked at the lesion itself and described it to be a part of the submandibular gland. The thyroid and parathyroids had a normal aspect.

The treating physician requested a third echography, which was performed by a different examiner, in order to clarify the diagnosis. This third ultrasound examination described an adenopathic block of 24/27 mm, with a rich vascularization. Computed tomography examination was recommended and it found a very large right submandibular gland that was hypodense compared to the left one on native examination (Figure 2). The right submandibular gland measured 2.9/2.4/4 cm anteroposterior/transversal/craniocaudal diameters, compared to 2.6/1.4/2.8 cm left submandibular gland. After intravenous contrast was administered, the right submandibular gland contrast uptake was more intense than that of the left one. The central Wharton duct was mildly dilated up to 2.3 mm inside the gland and was extremely dilated, up to 6 mm, in its submandibular portion; on its anterior extremity there was l large calculus, of 8/5 mm (anteroposterior/transversal diameters). The duct had iodophilic walls due to inflammation. There were some right submandibular inflammatory adenopathies with a short diameter of up to 1 cm, left submandibular adenopathies up to 7 mm, middle and superior jugular adenopathies up to 7 mm, and no other cervical abnormalities. No mediastinal, axillary and retroperitoneal adenopathies were found.

Oromaxilofacial surgical examination described oral floor abnormalities and purulent secretion at the site of the Wharton duct opening, which, together with the computed tomography examination, were suggestive for acute right sialadenitis with Wharton duct sialolith diagnosis (Figure 3). The surgeon performed a sialolithotomy and extracted a 10-mm yellow calculus (Figure 4). The procedure was followed by antibiotic therapy to prevent oral floor bacterial multiplication. Oral nonsteroidal anti-inflammatories and silagog alimentation were also recommended. Postoperative case evolution was favorable. The pH of the patient’s saliva was not measured because we did not have the tools to do that in our pediatric hospital.

## 3. Literature Review Methodology

Using the Medical Subject Headings MeSH term “pediatric sialolithiasis” and “pediatric sialolith”, we performed a Scopus and PubMed literature search for randomized controlled trials (RCTs), systematic reviews, observational studies, series of cases, and case reports from 1930 to 2020, in the English and Romanian languages only. Reports available only as abstracts were not taken into consideration. Studies reporting multiple sialolithiasis sites and studies also including adult sialolithiasis pathology were also included. We assessed gender repartition, epidemiology, presentation, diagnosis methodology, and treatment options in the 33 studies the search revealed.

## 4. Discussion

Sialolithiasis is the most common disorder of major salivary glands in adults and is the result of a deposition of calcium salts around a core made of desquamated epithelial cells, foreign bodies, bacteria, or mucus. Although alcohol consumption is not clearly associated with increased risk of salivary gland calculi, smoking is a well-recognized risk factor for this pathology [20]. Our patient did not have a history of active or passive smoking or alcohol consumption, and was not receiving chronic diuretic treatment, this type of treatment being another risk factor for calculi formation [21]. There was no history of antibiotic abuse, also a presumed risk factor [22], no cholelithiasis or nephrolithiasis, although some studies described an association between these pathologies and sialolithiasis [23]. No other presumed risk factors, such as diabetes, thyroid dysfunction, rheumatoid arthritis, Sjogren’s syndrome, epilepsy, hepatitis, or cirrhosis [22] were present in our case.

Due to the fact that two echography studies misinterpreted the lesion for an adenopathy, the path to the correct diagnosis was difficult. Literature also describes sources of errors, such as intravascular thrombi of the lingual veins, atherosclerosis of the lingual artery, calcified limphadenopaties, or pseudotumors [24,25]. In children, among the most common tumor formations in the cervical region are vascular malformations and infantile hemangiomas, the differential diagnosis sometimes requiring imaging investigations or biopsy and histological examination [26].

Submandibular “adenopathy” requires differential diagnosis with reactive adenopathies during viral or bacterial infections, tuberculosis, lymphoma, and acute lymphoblastic leukemia, autoimmune diseases, Kikuchi disease, and so on [27] (Table 1).

In children younger than 12, anterior cervical lymph nodes are the most frequently involved (38–45%), followed by the axillary and inguinal groups. Supraclavicular palpable adenopathies are most often found in children with certain types of cancer, whose tumors can be located in the thorax or in the abdomen [3].

Furthermore, plexiform neurofibromas of Recklinghausen disease should be differentiated from adenopathies. These lesions are sometimes distributed along the peripheral nerves and can sometimes overlap with lymph node projection areas. They are small, rubber-like lesions, covered by lilac skin [28].

In order to validate the hypothesis that there is a link between sialolithiasis and nephrolithiasis, Lustman et al. surveyed 245 sialolithiasis patients between 1968 and 1988 in the oro-maxilo-facial department of Hadassah Dental Medicine School [7]. Among these, 121 were men (49.4%) and 124 were women (50.6%). Their age varied between 6 and 94. Sialolithiasis was rare in the first decade of life (2.9%).

These patients complained about swelling (94%), pain (65.2%), purulent secretion (15.5%), fever (6%), while 2.4% were asymptomatic. Symptoms were worsened by eating and 80% of patients had frequent or continuous symptoms during the week prior to admission. Of the patients who participated in this study, only 1.2% of patients had both submandibular glands involved, 51.1% had right submandibular gland sialolithiasis, and 47.7% had left involvement only. In children, sialoliths are often located distally and can be observed by bimanual palpation [11]. In the cited study, 50.3% of calculi were located in the anterior portion of the Wharton duct, 18.7% in its posterior portion and 31% in the glandular hilum or the gland itself. Of the calculi located in the gland, 37% are close to the hilum, 43% are in the proximal canal system, and 20% are in the distal duct system or close to the sublingual papilla [29,30]. In total, 10% of sialoliths are inside the gland parenchyma [30,31].

In parotid sialolithiasis cases, most sialoliths (83%) are located in the distal portion of Stenton duct, while 17% are in the intraparenchimatous duct system [29]. Most parotid calculi (51%) are small (between 1 and 5 mm) [7]. The relatively larger diameter of Stenton’s duct could partially explain this difference [32]. Rauch et al. noticed multiple calculi in 16.9% of patients [5,33], while New et al. [34] describe recurrent sialolithiasis in the same gland in 8.9% of their patients, which may pose differential diagnosis difficulties with a frequent pediatric pathology, the endemic parotiditis.

Several studies [7,29] observed tumefaction to be the most frequent symptom, followed by pain [25], with local pain being the sole symptom in only 3% of patients [29]. Nahlieli et al. surveyed 146 pediatric patients aged 6 months to 17 between 2002 and 2016. They concluded that tumefaction and pain closely related to meals in sialolithiasis patients can be explained by increased intraglandular pressure caused by increased salivary secretion in the obstructed gland [29]. Levy describes this correlation in 43.9% of patients, as opposed to 72.4% in Maresh’s study [35].

In Nahlieli’s pediatric study tumefaction was the most frequent complain, as in our case, often accompanied by pain and less often by redness. Pain was worse after meals in 92% of children. In this group, 78% of submandibular calculi were diagnosed by bimanual palpation [29].

Regarding symptom duration, in Lustman’s study 76.6% of patients were symptomatic for over 1 year before receiving treatment, compared to 27.8% in Levy’s study [35] and 39% in New’s study [34].

The best imaging diagnosis method is considered to be intraoral X-ray, followed by orthopantomography [7,26,31]. Using these methods, sialoliths can be identified in 94.7% of cases. If an extraoral X-ray is used, submandibular and parotid calculi can overlap with bone structures or teeth.

In another study by Sigismund, 2959 calculi were identified by ultrasonography in 2322 patients [36]. With the help of ultrasonography, 80.4% were identified with localization in the submandibular gland. [29]; 83.1% of patients had a single sialolith, while 16.9% had multiple salivary calculi. The median length of a calculus measured by echo was 7.9 mm. In another study conducted between 2002 and 2016 [19], 70% of salivary calculi were identified by echography. Ultrasound imaging combined with bimanual palpation increased detection rate to 89% (32 out of 36 calculi). Overall, 4 calculi [11] were only diagnosed by sialadenoscopy. Other imaging methods, such as magnetic resonance imaging, computed tomography, and fine needle aspiration can be useful in some cases [19,34].

The preferred treatment is to remove the calculus obstructing the Wharton duct by oral approach. Levi et al. [35] state that 22% of parotid sialoliths can be removed using an endoscopic procedure. In some cases, this method could be applied to submandibular gland hilum sialoliths. Small calculi located close to the ductal orifice could be removed using a tear probe. Intraglandular sialoliths require submandibular sialadenectomy or partial parotidectomy [7,37]. The relatively small dimensions and distal localization of ductal sialoliths are important favorable prognostic factors. Only 5% of submandibular gland calculi can benefit from endoscopic treatment, most cases requiring a combined transoral approach [29].

Therapeutic management depends on sialolith localization. In children, the approach has shifted from conservative to more invasive treatment, as happened in our case [19]. Endoscopy can be performed for mobile calculi that are smaller than 5 mm [35]. Transoral surgical removal is adequate for treating persistent duct calculi. Shock wave extracorporeal lithotripsy (ESWL), performed in 14% of cases in Maresh’s study, is indicated in cases where the calculus cannot be visualized by endoscopy because it is located inside the glandular parenchyma or is embedded in the duct wall. In this study, sub-mandibulectomy was necessary in only one case. This procedure should be a last resort [35]. The calculus can be removed using an intraoral surgical procedure for most cases and glandular removal should be avoided as often as possible because of age and aesthetic concerns. Moreover, cases of spontaneous displacement were reported, submandibular gland massage could prove therapeutic if the calculus is located close to the sublingual papilla [38,39]. Other sialolith removal methods, such as endoscopy or ESWL can also be attempted. They can lead to full recovery or at least symptom improvement [40].

Gellrich’s study found no correlation between sialolithiasis and systemic diseases. In this study, nephrolithiasis coexisted with sialolithiasis in 10.7% of cases, a much higher percentage than that reported by other studies [5,41]. Gellrich followed up on patients for a mean of 3.4 years (minimum 9 months, maximum 12.3 years). A total of 78% of patients presented for follow-up evaluations. The team noticed the absence of symptoms immediately after treatment or at least significant pain improvement after treatment [19]. Other studies [5] described the same. A surgical approach should be decided based on sialolith dimension and localization [4,13,32,33].

The diagnosis of sialadenitis is a routine one for the oro-maxilo-facial surgeon, with a mean of 1.1 consultations. In sialolithiasis cases, however, a mean of 2.16 consultations can be necessary until the correct diagnosis is established, which can prolong disease duration by 5 months [18].

## 5. Conclusions

In our case, the diagnosis of sialolithiasis was difficult to establish, as multiple ultrasound examinations indicated that the palpable lesion was an adenopathy. Finally, a computed CT examination revealed the proper diagnosis. Although submandibular adenopathies are much more frequent in the pediatric age group, when faced with a firm, immobile submandibular lesion, the pediatrician should consider the sialolithiasis diagnosis. In this age group, the clinician and the radiologist should maintain a high suspicion index in order to correctly diagnose sialolithiasis and to do a correct differential diagnosis of pseudotumoral lesions located in this area.

## Figures and Tables

**Figure 1 diagnostics-12-00792-f001:**
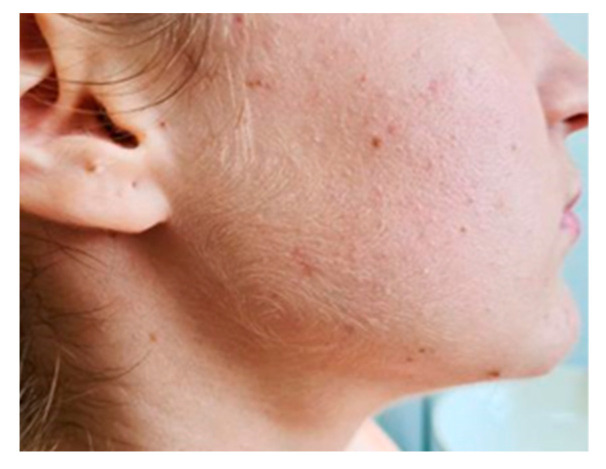
Submandibular pseudotumoral lesion.

**Figure 2 diagnostics-12-00792-f002:**
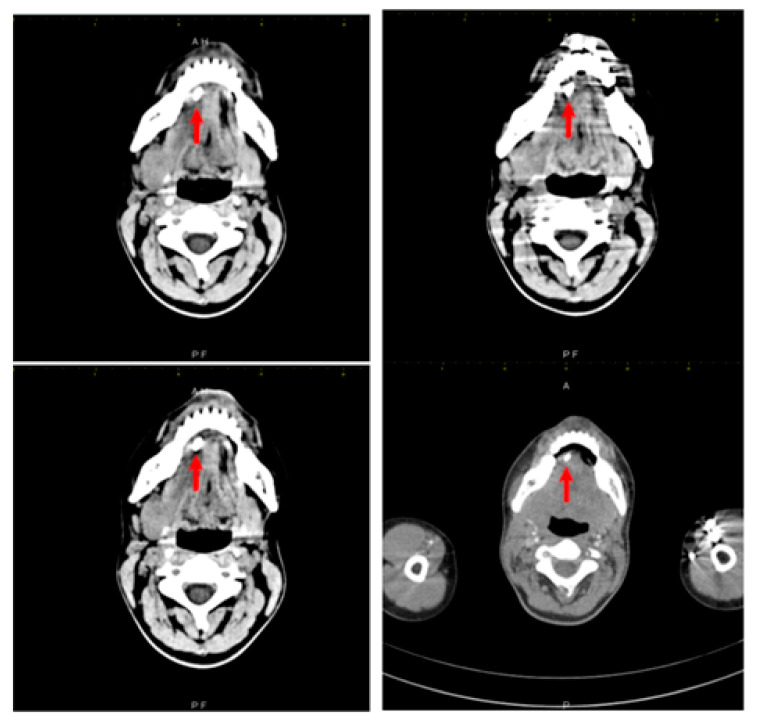
Right submandibular sialolithiasis, various CT scan sections.

**Figure 3 diagnostics-12-00792-f003:**
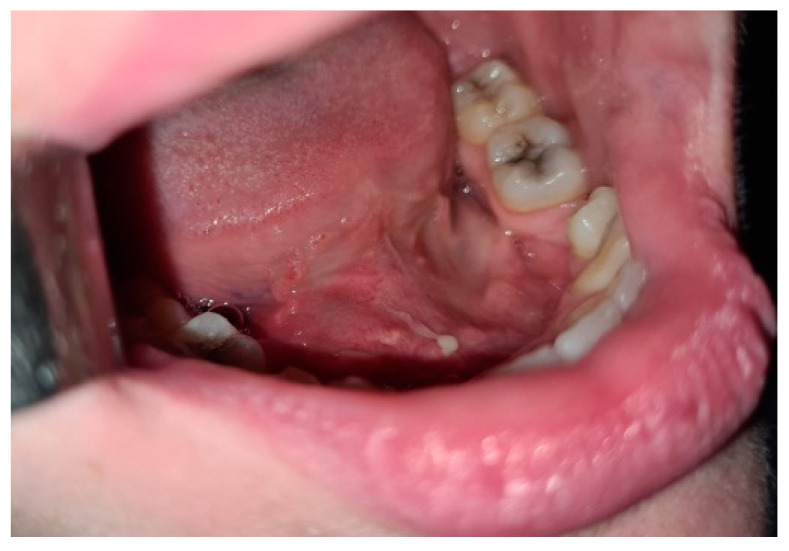
Intraoral view-oral floor bulging, purulent secretion at the Wharton duct orifice.

**Figure 4 diagnostics-12-00792-f004:**
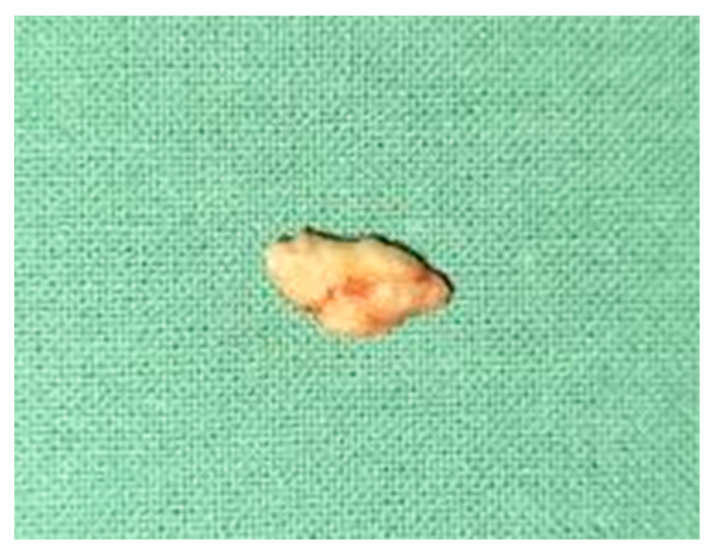
Extracted Wharton duct calculus.

**Table 1 diagnostics-12-00792-t001:** Adenopathy causes in children.

Type	Subtype	Diagnostic Traits
Infectious causes	Viral causes: Citomegalovirus, Epstein–Barr virus, HIV, mumps, rubella Bacterial causes: group A Streptococus pyogenes, Staphilococcus aureus, Tularemia, Brucellosis, Bartonela henslae Fungi: Cryptococcus, Aspergilus, Hystoplasmosis, Coccidiomycosis Mycobacteria: M. tuberculosis, M. avium Protozoa: Toxoplasma, Malaria	- fever, suppuration, fistulisation - positive serology, several firm, puss containing lymph nodes - immunocompromised patient - positive anamnesis, fever may or may not be present, positive skin prick test, abnormal X-ray - positive serology
Malignancies	Leukemias, lymphomas Solid tumors: neuroblastoma, rhabdomiosarcoma, carcinoma	- generalized adenopathies, systemic symptoms - isolated adenopathies posterior to the sternocleidomastoidian muscle, supraclavicular, larger than 3 cm, firm, non-painful, rapidly growing
Autoimmune diseases	Juvenile idiopathic arthritis Systemic lupus erithematosus	- articular involvement, positive rheumatoid factor - skin rash, arthritis, anemia, neuropsihical abnormalities, antinuclear/anti double strand antibodies
Histiocitosis	Malignant histiocytosis Langerhans histiocytosis, hemophagocitic syndromes	- histopathological diagnosis
Stocking diseases	Niemann Pick disease Gaucher disease	- genetic diseases implying neurological manifestations, hepato-splenomegaly and bone deformations
Others	Sarcoidosis Kikuchi disease Castelman disease	- benign lymph node involvement - mediastinal and pulmonary lymph node infiltrates
Vaccinations Immunodeficiencies	Variole, tuberculosis Chronic granulomatous disease	- recent vaccination - recurrent infections history - hepatosplenomegaly - granulomas

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
