# Peer review of "Multiple Faces of Cervical Lesions in Children"

_diagnostics, 2022, doi:10.3390/diagnostics12040792_

Round 1

Reviewer 1 Report

The Authors described a very valuable case of a patient with a rare disease in this age group. However, there were a few inaccuracies in the manuscript. Below is a list of comments:

Line 47 - What were the age of the children with Sialolithiasis in the cited publications?

The authors mention that one of the factors contributing to the formation of sialolith is the pH of the saliva and the concentration of calcium (line 59). So why was the patient's pH not measured? Obviously, measuring the concentration of calcium ions in saliva is more difficult and performed infrequently, and most of all, not routinely.

Was the saliva pH determined in a patient after the correct diagnosis?

Line 141 - was the patient's exposure to passive smoking also determined in the interview?

Lines 170 - 171 - the cited study was conducted on a group of people aged between 6 and 90. However, the given percentage relates to children in the first decade of life, the described patient is in the second decade. Therefore, it would be more logical to quote the frequency of the disease in this age group.

Lines 184-189 - a case report of parotid sialolithiasis adds nothing to the description of the patient's case and can, in principle, be omitted.

The authors should indicate why the X-ray examination was not performed, since the literature data indicate that it is the most useful (line 204)

Line 208 - in many places the manuscript describes the usefulness of a given method to confirm the diagnosis as a percentage. The number of cases is given here (2959), but there is no indication of the total number of people tested. As a result, it is difficult to assess the usefulness of ultrasonography in the diagnosis of the disease under study.

Author Response

Response to reviewer 1:

Thank you for your kind comments and advices!

We addressed the deficiencies you kindly pointed out as follows:

  1. Line 47 - What were the age of the children with Sialolithiasis in the cited publications?

The age is between 4 and 16 years, with an average of 10.4 years, we added this in the text.

  1. The authors mention that one of the factors contributing to the formation of sialolith is the pH of the saliva and the concentration of calcium (line 59). So why was the patient's pH not measured? Obviously, measuring the concentration of calcium ions in saliva is more difficult and performed infrequently, and most of all, not routinely. Was the saliva pH determined in a patient after the correct diagnosis?

The pH of the patient’s saliva was not measured because we did not have the tools to do that in our pediatric hospital. We added this to the text.

  1. Line 141 - was the patient's exposure to passive smoking also determined in the interview?

The patient was neither an active nor a passive smoker; we inserted this in the text.

  1. Lines 170 - 171 - the cited study was conducted on a group of people aged between 6 and 90. However, the given percentage relates to children in the first decade of life, the described patient is in the second decade. Therefore, it would be more logical to quote the frequency of the disease in this age group.

The cited article does not mention the frequency of the disease in the second decade of life.

  1. Lines 184-189 - a case report of parotid sialolithiasis adds nothing to the description of the patient's case and can, in principle, be omitted.

If possible, we would like to keep these lines due to sometimes difficult differential diagnosis with endemic parotiditis, a condition that is much more frequent in the pediatric population. We added a remark on this in the text.

  1. The authors should indicate why the X-ray examination was not performed, since the literature data indicate that it is the most useful (line 204)

The X-ray examination was not performed because we followed the indications of our radiologist who was confused at that time about the diagnosis. She thought what she saw was an adenopathic block, suggestive for a malignant process, and she recommended a CT examination instead of a simple X-ray.

  1. Line 208 - in many places the manuscript describes the usefulness of a given method to confirm the diagnosis as a percentage. The number of cases is given here (2959), but there is no indication of the total number of people tested. As a result, it is difficult to assess the usefulness of ultrasonography in the diagnosis of the disease under study.

The number of individuals tested was 2322. We included this in the text.

Also, we zoomed in on the picture, to make the patient harder to recognize, while still keeping the area of interest.

Reviewer 2 Report

Dear Authors, I was pleased to review this interesting case report. This is the presentation of a clinical case of a 16-year-old patient with swelling, pain whose correct diagnosis of sialolithiasis was made after some ultrasound examinations (3). I believe that the paper is well written and clear, as well as considering that we are talking about a rare disease in the pediatric / juvenile population. In the Discussion section we refer to the review of the literature that had Scopus and Pubmed as a reference database. Here are my suggestions to the authors: 1. Check carefully the data of the works cited, because there are errors in the transcription (percentages) probably typing.

2. 

Minor comments:

Line 60: please, correct: [80-92%).

Line 87: Ebstein is incorrect, please correct in Epstein.

Line 88-89: please, the names of the infectious agents all in capital letters.

Line 115: puss is wrong. Please correct.

Line 144: 23]. Please add a square bracket

Table 1: Please, arrange the alignments between the various columns

The style of the references is not that of MDPI: it is important before submitting an article. Please correct the references.

Author Response

Reviewer 2

Dear Authors, I was pleased to review this interesting case report. This is the presentation of a clinical case of a 16-year-old patient with swelling, pain whose correct diagnosis of sialolithiasis was made after some ultrasound examinations (3). I believe that the paper is well written and clear, as well as considering that we are talking about a rare disease in the pediatric / juvenile population. In the Discussion section we refer to the review of the literature that had Scopus and Pubmed as a reference database. Here are my suggestions to the authors:

  1. Check carefully the data of the works cited, because there are errors in the transcription (percentages) probably typing.
  2.  

Minor comments:

Line 60: please, correct: [80-92%).

Line 87: Ebstein is incorrect, please correct in Epstein.

Line 88-89: please, the names of the infectious agents all in capital letters.

Line 115: puss is wrong. Please correct.

Line 144: 23]. Please add a square bracket

Table 1: Please, arrange the alignments between the various columns

The style of the references is not that of MDPI: it is important before submitting an article. Please correct the references.

Response to reviewer 2:

Thank you for your kind comments and advices!

Indeed, there were some mistakes in percentage transcription, we fixed them. We also addressed the other typos you pointed out; we formatted the table and corrected the references.

Also, we zoomed in on the picture, to make the patient harder to recognize, while still keeping the area of interest.